# Short-Term Prediction of Energy Consumption in Demand Response for Blocks of Buildings: DR-BoB Approach

**Nashwan Dawood**

School of Computing, Engineering and Digital Technologies, Teesside University, Middlesbrough TS1 3BA, UK; n.n.dawood@tees.ac.uk; Tel.: +44-1642-342405

**Abstract:** Load forecasting plays a major role in determining the prices of the energy supplied to end customers. An accurate prediction is vital for the energy companies, especially when it comes to the baseline calculations that are used to predict the energy load. In this paper, an accurate short-term prediction using the Exponentially Weighted Extended Recursive Least Square (EWE-RLS) algorithm based upon a standard Kalman filter is implemented to predict the energy load for blocks of buildings in a large-scale for four different European pilot sites. A new software tool, namely Local Energy Manager (LEM), is developed to implement the RLS algorithm and predict the forecast for energy demand a day ahead with a regular meter frequency of a quarter of an hour. The EWE-RLS algorithm is used to develop the LEM in demand response for blocks of buildings (DR-BOB), this is part of a large-scale H2020 EU project with the aim to generate the energy baselines during and after running demand response (DR) events. This is achieved in order to evaluate and measure the energy reduction as compared with historical data to demonstrate the environmental and economic benefits of DR. The energy baselines are generated based on different market scenarios, different temperature, and energy meter files with three different levels of asset, building, and a whole pilot site level. The prediction results obtained from the Mean Absolute Percentage Error (MAPE) offer a 5.1% high degree of accuracy and stability at a UK pilot site level compared to the asset and whole building scenarios, where it shows a very acceptable prediction accuracy of 10.7% and 19.6% respectively.

**Keywords:** demand response; energy baseline; Extended Recursive Least Squares Algorithm; energy load forecasting

## 1. Introduction

An energy baseline profile is a reference which allows a comparison between the energy performance before and after a change made to the site or system in a demand response (DR) situation. The accuracy of the baseline calculations is important to identify energy reduction in buildings. The baseline establishes the "before" by capturing a site or a system's total energy use prior to making improvements and changes [1]. It accounts for energy-affecting factors like temperature or production volume.

In the literature, many baseline forecasting models have been developed during recent years to predict the energy demand and energy load forecasting based on different techniques. The supply industry requires forecasts with a model horizon that ranges from a short-term cover up to days ahead, or the long-term that is able to cover years ahead. As the load forecasting plays a major role in determining the prices of the energy supplied to customers, an accurate prediction is vital for the energy companies, especially when it comes to the baseline calculations that are used to predict the consumption load.

The development of an energy baseline to evaluate and measure the energy efficiency is not always straightforward and is usually a challenging task. The baseline definition should distinguish between any changes in energy consumption that are caused by applying a demand response event or any energy consumption measures that are caused by other reasons and factors (e.g., weather). This introduces problems sometimes, particularly when the energy service of the specific application has changed during the measurements' implementation or when some of the factors cannot be controlled. Understanding these issues helps to detect and quantify the effects of the applied event on the energy consumption changes. In general, energy companies do not publish the details of their energy baseline calculations. They do, however, sometimes explain the selection process used by their baselines, which is the case for the calculations used by Pennsylvania, New Jersey, Maryland Interconnection LLC (PJM), KIWI power, The California Independent System Operator (CAISO), and KIBER-net project [2]. In addition to the forecast model itself, the application, the market, and the amount of historical data used to generate the baseline in each model varies from one research to another. Many researchers used neural networks, self-organizing neural (SON) networks, artificial intelligence (AI) [3–6], and fuzzy neural models [7–10] as promising techniques to predict the energy load. In recent times, much research has been carried out on a Recursive Least Square (RLS) algorithm, which is an adaptive filter algorithm that recursively finds the coefficients that minimize a weighted linear least squares cost function relating to the input signals, along with the Kalman Filtering algorithm [5,11–16]. The RLS algorithm has been widely used to increase the accuracy of the load forecasting. Kissock et al. [15] presented a general method for measuring plant-wide industrial energy savings that takes into account changing weather and production between the pre and post retrofit periods. The method incorporated Least Squares Regression model with additional search techniques but was implemented in a simple way using data analysis software. One of the most important limitation of Kissock's study is in the system domain, where the method attempts to determine savings from individual subsystems using whole-plant energy use.

Dotzauer [16] considered a method for predicting the heat demand in two district heating systems (large and small). The author used a load predictor model within the Linear Least Square optimization equation. A commercial prediction software called Aiolos was also used in parallel in the Scandinavian Electricity-spot market. The simple model was based on the insight that the load is mainly affected by the outdoor temperature and the social behaviour of the consumers. The yearly variation was considered using historical data for 6 weeks available before the forecast horizon that was taken from the corresponding period in previous years. Dotzauer's model's accuracy varied based on two things: the size of the system (large and small) and the period of the year (divided into four periods). The overall error from the study results was found to be between 6.51% and 15.22%.

The work in [17–19] used combined previous models of neural networks, Kalman Filtering, and RLS techniques together in different levels to predict the load forecasting.

Predicting the future energy demand is important because this demand is basically unknown, and therefore different techniques and strategies have been proposed for this reason. The techniques included in any optimization work should be accurate and reliable enough to cover the prediction horizon ahead.

Wang et al. [20] proposed a combined heat and power (CHP) based district heating (DH) system with renewable energy systems (RES) and an energy storage system (ESS). They developed a modelling and optimization method to minimize the overall costs of the net acquisition for heat and power in deregulated power markets, but with a longer length of the planning horizon employed compared to other similar studies. The results of Wang's study assumed both heat and electricity demand were explicitly and accurately known. Nevertheless, the conclusions indicated that amongst other things, heat storage is utilized in proportion to the level of the daily load variability.

Gruber et al. [21] presented the application of an advanced building energy management tool for an emulated medium size hotel where the resulting minimization problems were formulated as Mixed Integer Linear Programming (MILP). Gruber's study had a major difference over other studies

in that Gruber's building energy management optimized the use of the energy system resources for the next 24 h in two successive steps. In the first step, the approach solved a medium-term optimization problem over a horizon of one day with a sampling time of one hour. The experimental results from Gruber's study showed that the proposed strategy increased energy efficiency, reduced energy costs, and increased the quality of the optimization through a high level of accuracy. The building energy management results also showed the possibilities to validate complex strategies under realistic conditions using power-hardware-in-the-loop techniques.

Ommen et al. [22] investigated the impact on operational management of the combined utility technologies for electric power and district heating (DH) of Eastern Denmark. Ommen's work utilized three different optimization approaches of linear programming (LP) which is used as a benchmark, LP with binary operation constraints, while the third approach used nonlinear programming (NLP). Results from the three different optimization types of linear programming (LP), mixedinteger programming (MIP), and Nonlinear programming (NLP) were that the LP has the lowest amount of constraints of the three. However, comparisons of the derived operation of the units showed significant differences between the three methods. For the heat pumps, the linear optimization yielded the lowest amount of operating hours. While using MIP optimization, the number of operational hours for heat pumps was increased by 23%, while for an NLP optimization the increase was more than 39%. Considering the total production of heat from the heat pumps, the two approaches differed by approximately 23% and 32%, respectively.

Chen et al. [23] proposed and tested a Support Vector Regression (SVR) model to calculate the demand response baseline for office buildings. The author used 8 hours' prediction horizon for only working days based on historical data. Wang et al. [24] established short-term prediction models based on a regression algorithm that was used to predict the power consumption and validate it using real time monitoring data. The prediction models were validated to have great accuracy and general applicability for the whole week horizon ahead, which provides a good foundation for the large-scale buildings in diagnosing, energy monitoring, energy saving reforms, and regulatory work. Bo-Juen et al. [25] proposed a new learning technique based on a Support Vector Machine (SVM) model for load forecast. The author developed an SVM model based on Least Square Regression (LSR) as a learning technique. Half an hour (30 min) was recorded as frequency data. The prediction works for a month ahead using 2 years of load demand data and 4 years of temperature data, while 6 months of historical data are used as training base.

Ferdyn-Grygierek et al. [26] developed a multivariant analysis for the determination of the accuracy of the seasonal heat demand in buildings. A linear regression method was used to analyse data obtained during short periods of measurement. The analyses were carried out using TRNSYS, ESP-r, and CONTAM programs, and the numerical models of the multifamily building and school building were used for the simulation. The simulations were performed using the TRNSYS, ESP-r, and CONTAM programs. The paper identified the impact of variables such as airtightness, insulation, and occupancy schedule. The main conclusions of the paper were that for better installed buildings, the uncertainty of estimating the heat demand increased in relation to a building with less insulation on the external partitions, and the uncertainty of estimating the heat demand increased in relation to a building with less airtight windows.

The other important issue is the measure used to calculate the prediction accuracy of a load forecasting model. Many formulas and techniques have been used for this purpose in the literature. Mean Percentage Error (MPE), Mean Absolute Scaled Error (MASE), Mean Directional Accuracy (MDA), Mean Arctangent Absolute Percentage Error (MAAPE), and the measure used in this work, namely Mean Absolute Percent Error (MAPE). These measures have different formulas, but the results of their measures are quite similar, with almost the same level of accuracy.

In this paper, we present mathematical models and software solution for short-term load prediction, which uses an Exponentially Weighted Extended Recursive Least Square (EWE-RLS) algorithm in a similar approach to the work of Short [17]. First, by extending the prediction algorithm to pick up

demand response (DR) signals, if present, to enable the DR baseline calculation. Second, to implement the prediction model as a component of Local Energy Manager (LEM) deployed in DR-BoB pilot demonstration sites to provide heat and electricity demand forecasts, in addition to LEM's optimization function, which is neglected in this paper. The validation of the generated DR baseline for all meters demonstrates the effectiveness of the prediction algorithm, and the implementation results indicate that the generated baselines were accurate with less percentage errors of between 6% and 16%, although some differences in MAPE are evident for all meters located at different levels. The architecture of DR-BoB and the interoperability of its technological tools are also described in this paper.

The remainder of this paper is structured as follows. Section 2 presents the contributions in this work and briefly highlights the developed software tool scheme. Section 3 provides an overview of the framework used for calculating energy savings. Section 4 discusses the baseline generation along with the baseline validation and assessment results. Finally, the paper concludes in Section 4.

## 2. Contribution

Demand response in blocks of buildings (DR-BOB) is a H2020 large-scale project. An accurate short-term prediction using the Exponentially Weighted Extended Recursive Least Square (EWE-RLS) algorithm based upon a standard Kalman filtering has been implemented to predict the energy load forecasting during different demand response scenarios for blocks of buildings in four European pilot sites. A new software tool, namely Local Energy Manager (LEM), has been developed to implement the extended version of the RLS algorithm and MAPE measurement to predict the forecast for energy demand a day ahead with a regular meter frequency of quarter an hour.

Figure 1 illustrates the architecture of the end-stage deployment model for DR-BOB with the LEM in different pilot sites of the DR-BOB project. This is how the technical solution looks like when each of the elements are implemented at all of the pilot sites. It does not show the specific equipment that the LEM interfaces with at each of the sites, as although this is part of the integration of the equipment itself, it is outside the scope of this study. The baseline results of this work have been evaluated at different stages to measure the accuracy of the load forecasting prediction a day ahead, assessed by Siemens, and the $CO_2$ reduction and the economic benefits have been analysed and evaluated by Scientific and Technical Centre for Building (CSTB) in France.

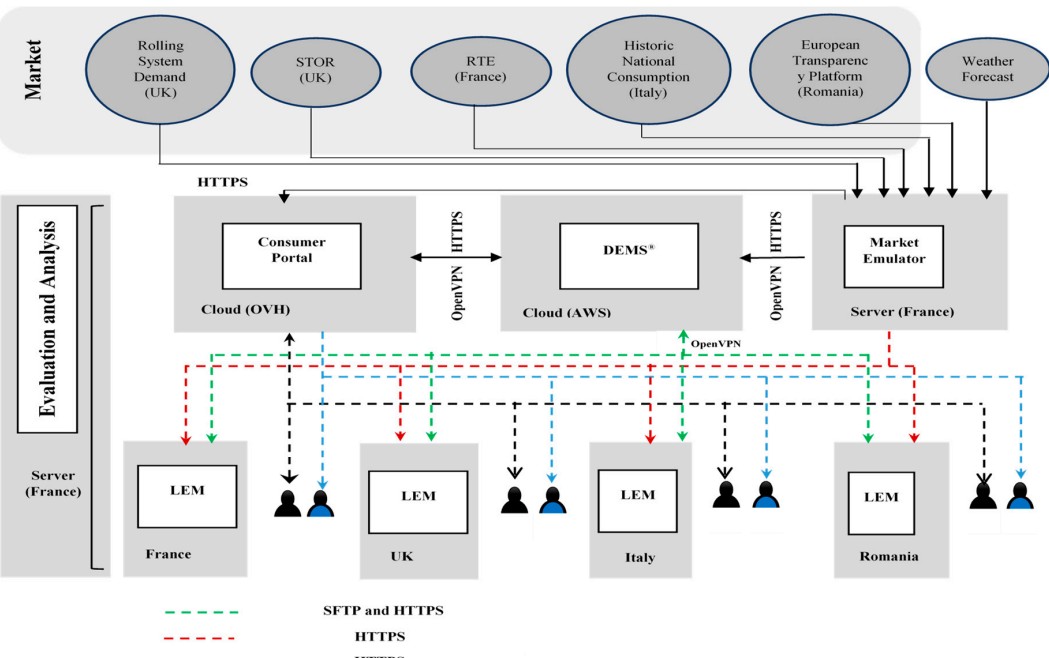

**Figure 1.** The demand response for blocks of buildings (DR-BOB) Deployment Model [27].

Outliers in sensor networks are those measurements that significantly deviate from the normal pattern of the sensed data. Depending on their nature, the outliers may not cause any major effect, but can cause considerable impact and contribute to the distortion of the distribution of the series during the data analysis. Depending on the purpose of the study or analysis, data may or may not need correction. Correction is usually needed if the failures are due to measurement problems. Generally speaking, there are many different causes of these outliers. Current and voltage transformer saturation, measuring or transduction equipment damaged, parameterization or improper installation, and registration problems in the database are some of them. Missing temperature or energy consumption data are the most common problems and cause a clear low spike in the data curve. If corrupted data are used for management and decision-making, then decisions might be inaccurate and inadequate. As a result, actions performed based on wrong data can be incorrect and can lead to undesirable consequences. The situation becomes especially critical in the baseline assessment scenario that requires a certain minimum level of predictive accuracy.

The data preprocessing procedure has been applied to the data used in this study to make sure outliers will not impact the generated baseline. The Linear Interpolation method has been implemented and used in this study to fill missing data and handle any zeros and negative values.

*2.1. Local Energy Management System (LEM)*

An LEM is a hardware and software solution developed and configured for each demonstration site. It interacts with the other elements as per the system architecture shown in Figure 2. The Market Emulator (ME) for the DR-BOB is standing in place of a Transmission System Operator, Distribution System Operator, or an aggregator, and generates demand response events. The Virtual Energy Plant (VEP) allocates these requests to the LEM at each pilot site, while the Consumer Portal (CP) manages interactions with the facilities manager and building occupants. The four demonstration sites have different uses, physical forms, and market and climatic contexts. Each block of buildings has a unique configuration of assets, metering, and management, largely encountered at the start of the project as Demand Response Technology Readiness Levels (DRTRL). The DRTRL assessment has been developed and applied to all demonstration sites. As such, all sites have required some degree of investment, predominantly in metering to be able to deliver DR-BOB functionality and measure impact effectively.

The LEM solution can adapt to fluctuations in the energy demand or production, subject to dynamic price tariffs and changing weather conditions. It establishes a baseline for specific assets' demand focusing on short-term forecasting of both heat and electrical loads, along with unit commitment scheduling and economic dispatch optimization. As part of this, the baseline establishes an accurate prediction strategy based on historical meter values from the main site Building Management System (BMS), utilization patterns, and weather functions. This forecast is more accurate as the window horizon shortens, i.e., for one day ahead forecast, the baseline has typically a 4%–20% Mean Absolute Prediction Error (MAPE) with the real demand of each asset in the LEM algorithm. However, this algorithm loses accuracy as the rolling horizon expands. The approach adopted builds on recent research employing Mixed Integer Linear Programming (MILP) models and nonlinear boiler efficiency curves and extends this work in a rolling horizon context.

The one day ahead rolling horizon is consistent with most of the requests for demand response actions to be taken. Therefore, this baseline would be convenient to be established as a way to examine the effectiveness of the DR actions in lowering, shedding, or shifting demand across assets within blocks of buildings. Part of the benefits of the evaluation and analysis work package is discerning the best approach to determine the effectiveness of the baselines in demand response programs.

The LEM, as shown in Figure 2, is the single point of integration for the site. It manages the local energy, taking into account patterns of consumption, local pricing, weather, and the behaviour of the citizens in the building. It is also a demand response router and protocol converter to allow communication between other devices or assets on pilot site. As such, one of the first aims of developing and using LEM is to manage, generate and identify the Demand Response Technology Readiness Level

for each site and the necessary modifications required to enable the DR-BOB LEM to generate the baseline for different energy meters and temperature files.

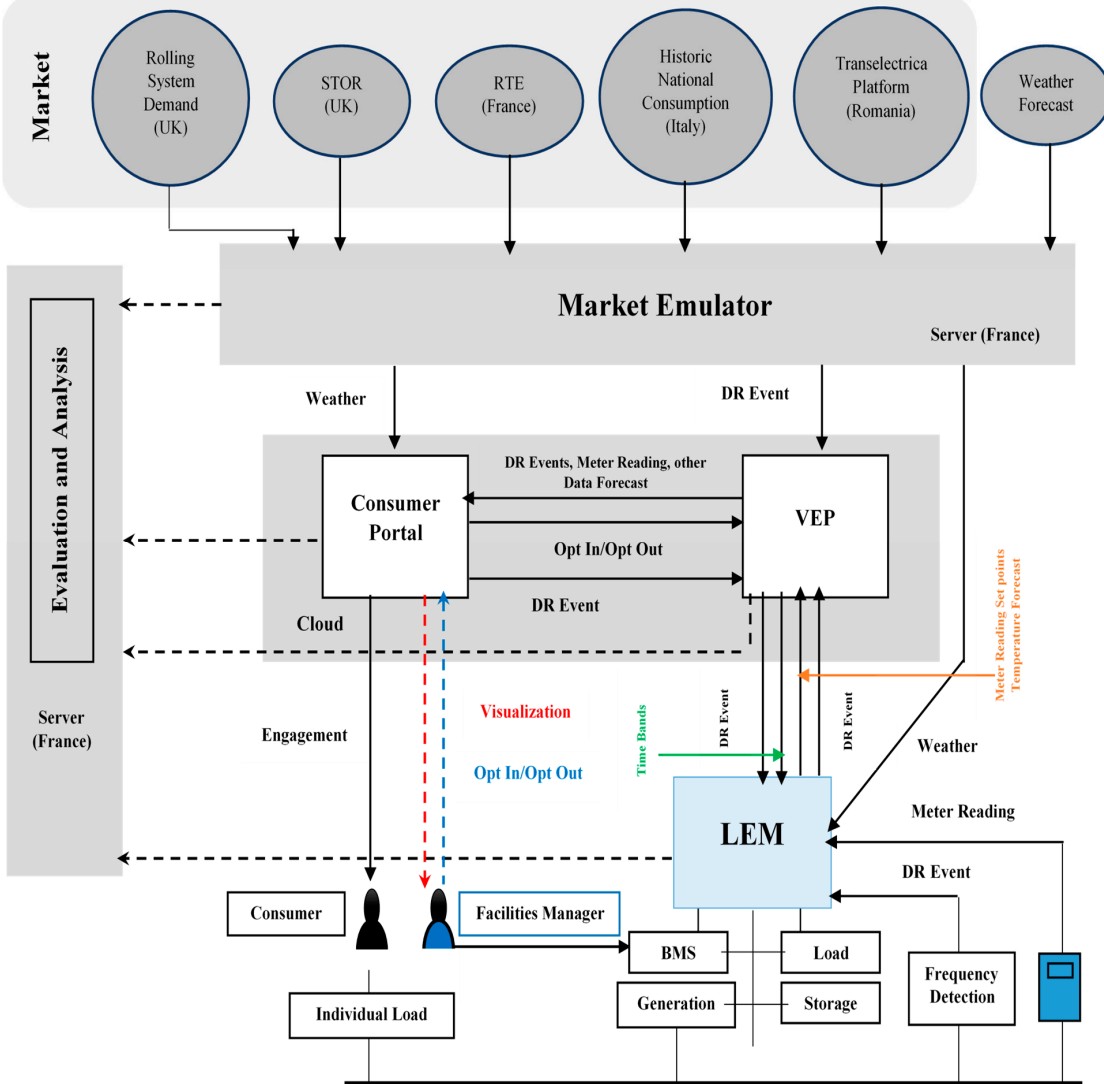

**Figure 2.** The Situation of the Local Energy Manager (LEM) and its Associated Interfaces within the DR-BOB Solution [28].

*2.2. Energy Demand Response and Demand Response Event Scenarios*

There are a number of data sets that are collected as standard (i.e., meter readings and temperature). There is also a general need to visualize the data and a standard process has been followed to achieve this. Data collection is an ongoing process and will occur continuously, but the presentation will be requested ad hoc by the Facilities Manager (i.e., Energy Manager).

Data collection is relatively simple and covers all forms, including readings, forecasts, and recommendations. The process is for the LEM to collect the readings from the appropriate site system, Building Management System (BMS), or meter data collection system and push it to the Virtual Energy Platform (VEP) where it is then ingested and made available to the Consumer Portal (CP). The ingestion process in the VEP runs every 15 min but could be changed to a lower or higher frequency if necessary. The standard process during the baseline generation in the DR-BOB runs every 15 min.

Figure 3 illustrates the standard event receipt and initiate processes for any DR events. It also shows the process of opting in or opting out. The two processes are closely linked, therefore these have

been combined in one sequence diagram. The opting in and out features allow the facilities teams to decide before an event begins to exclude assets, either automated or in a manual way. For example, perhaps there is a need to keep the heating, ventilation, and air conditioning (HVAC) running during a DR event; this feature ensures that the HVAC is not included in the LEM's control strategy.

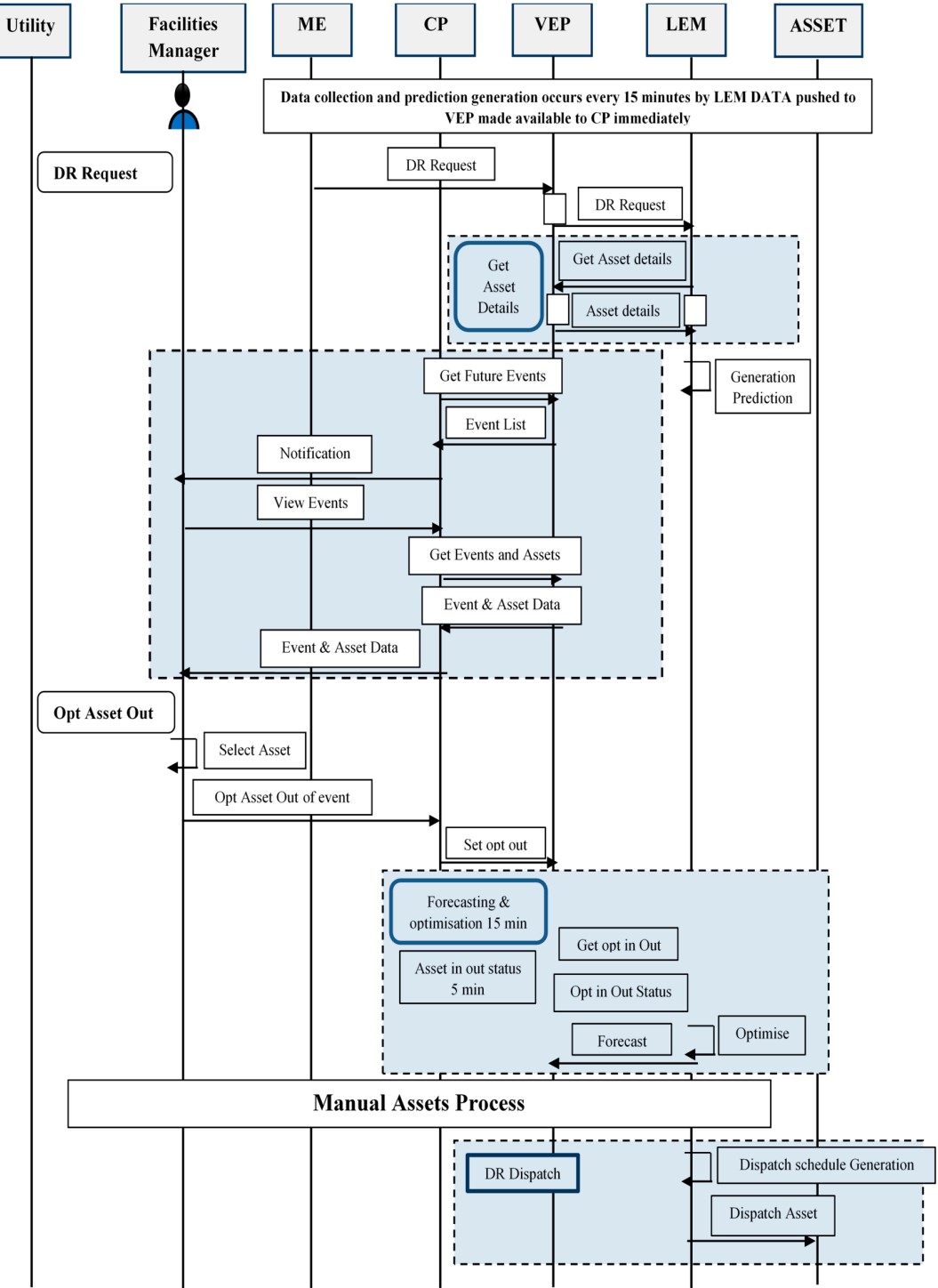

**Figure 3.** Standard event and opt in/opt out [28].

The process begins with the receipt of an event, this is illustrated as coming from the ME, but as has been mentioned previously it could also come from the LEM or CP. The first part of the sequence diagram shows how the events can be viewed in the CP.

As each subsystem performs actions in parallel, at first, the diagram may appear confusing. The following can provide a simple explanation. When an event is received by the VEP, it is immediately sent to the LEM, and made immediately available for the CP.

At the LEM:

- When the LEM receives the event, it looks up the details of the virtual assets in the VEP and begins to create the forecasts.
- The LEM continues to check the status of the virtual assets every five minutes.
- The LEM optimizes the forecasts and continues to make these available to the VEP.

In parallel at the CP:

- The facilities manager can look at the various sets of data made available from the VEP. These include, but are not limited to, the virtual assets, meters, and the channel data.
- The facilities manager can opt an asset out of an event at any point before the start of the DR event.

At the start time of the event, or before if there is a ramp up time for the asset:

- The LEM will generate and execute a dispatch schedule that will send control signals or will set the set-points.

### 2.3. Distributed Energy Management System (DEMS)

There is one instance of the Distributed Energy Management System (DEMS) which has been deployed to a single Linux Virtual Machine (VM) on the Amazon Web Services (AWS) cloud service. The DEMS is an existing Siemens product that has been implemented as part of the DR-BOB technical solution. DEMS' primary purpose is to enable common configuration of programs, virtual assets, meters, channels, baselines, time-bands, and more. DEMS is also used to act as a persistent store of readings and other collected data from the pilot sites via the LEM. As well as providing configuration and data storage, it is the responsibility of DEMS to route the DR events from any source to the right LEM configuration at the right demonstration site. DEMS facilitates the passing of recommendations generated by the LEM to the CP for dissemination to the appropriate Facilities Managers (FMs) and consumers. Finally, DEMS provides all this data and configuration to the Evaluation and Analysis (E&A) at regular intervals. The DEMS implementation for the DR-BOB project only makes use of standard elements of the data structure and existing processes, there is no customization. Each pilot site is set up as a single customer with multiple premises.

The interface's configuration required for each pilot site is as follows. The Market Emulator (ME) receives notification of demand response (DR) events. The Consumer Portal (CP) allows presentation of data to the consumer portal users and allows users (Facilities Managers) to opt out (make assets unavailable) and the LEM to push notifications of events and receive meter data. LEM and DEMS connectivity is via a secure VPN tunnel using OpenVPN technology. The benefit of using OpenVPN is its simple setup on the server and client side. LEM sends meter readings and meter reading forecasts every 15 min to DEMS in UAAV2 XML format using Secure File Transfer Protocol (SFTP). LEM polls for and receives DR events from DEMS in Open Automated Demand Response (OpenADR 2.0) format and over a proprietary webAPI in some circumstances. The OpenVPN connection is established first, then communication using the OpenADR 2.0 protocol can proceed.

### 2.4. Prediction Algorithm and Baseline Assumptions

This section describes the calculation methods for the defined Key Performance Indicators, and the mathematical model used in this work. The evaluation for the yearly energy consumptions, energy and $CO_2$ reduction, and the economic benefits help to understand and analyse the impact of the DR-BOB solution implementation.

2.4.1. Energy Savings Calculations

The calculations used for energy savings or over consumption corresponds to the reduction/increase of energy consumption in kWh during a whole DR event(s). This indicator is calculated for both electricity alone and for all types of energies used generally as well. This indicator and the difference between electricity import and electricity consumption are illustrated in Figure 4.

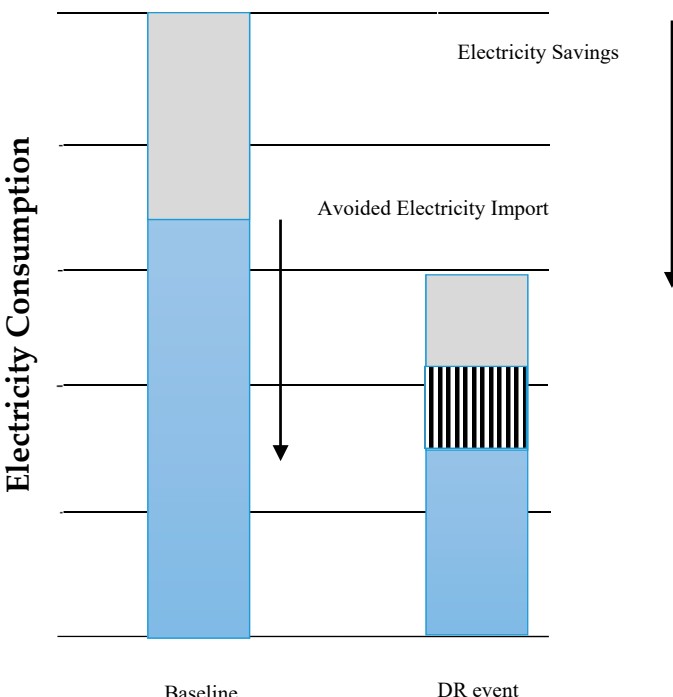

**Figure 4.** Illustration of Energy KPIs (Key Performance Indicators) and Volumes/Electricity.

The data required for the calculation are:

- $\delta_{shed+shift}$: DR event trigger ($\delta_{shed+shift} = 1$ during shedding and shift periods, and 0 elsewhere).
- $P_{DR}\ (t)$: asset real energy consumptions (for each energy vector) during DR event, in kW.
- $P_{baseline}\ (t)$: asset baseline energy consumptions (for each energy vector) without DR event, in kW.
- The calculated data will be:
- $E_{savings}$: Energy savings (negative in case or overconsumption).
- In kWh of primary energy.
- In % for all considered time and space perimeters.
- $E_{savings,elec}$: Electricity savings (negative in case or overconsumption).
- In kWh of final energy.
- In % for all considered time and space perimeters.

The electricity savings are calculated as the gap between DR scenario consumption and baseline consumption during both shedding and shift periods [29]:

$$E_{savings,elec}(\Delta t) = \int_{t\in\Delta t} \left(P_{baseline,elec}(t) - P_{DR,elec}(t)\right) \cdot \delta_{shed+shift}(t).dt \tag{1}$$

When working with discontinuous values, the approximation becomes:

$$E_{savings,elec}(\Delta t) \approx \sum_{t\in\Delta t} \left(\overline{P}_{baseline,elec}(t) - \overline{P}_{DR,elec}(t)\right) \cdot \delta_{shed+shift}(t) \tag{2}$$

The energy savings for all other energy vectors (fuels, district heating) are calculated in the same way. The global primary energy savings (or overconsumption) are:

$$E_{savings}(\Delta t) = \sum_{ev} E_{savings,ev}(\Delta t) \tag{3}$$

where *ev* refers to the related energy vectors. All savings need to be converted in kWh of primary energy (kWh$_p$). This conversion is done by considering:

- National electricity conversion factors (in kWh$_p$/kWh$_{elec}$).
- Local district heating conversion factor (in kWh$_p$/kWh$_{heat}$).
- Lower calorific value of different fuels (in kWh/m$^3$).

In order to convert these savings into a percentage, these volumes are divided by the baseline energy consumption for the considered space perimeter [28].

$$E_{savings}[\%] = 100 \frac{E_{savings}[kWh_p]}{E_{consumption,baseline}[kWh_p]} \tag{4}$$

With:

$$E_{consumption,baseline}(\Delta t) = \sum_{\forall ev}\left(\int_{\Delta t} P_{baseline,ev}(t).dt\right) \approx \sum_{\forall ev}\sum_{t\in\Delta t} \overline{P}_{baseline,ev}(t) \tag{5}$$

2.4.2. $CO_2$: Reduction of Greenhouse Gases Emissions

This indicator corresponds to the reduction of equivalent $CO_2$ emissions in kg $CO_2$eq due to the DR implementation.

The needed measures and information are:

- $D_{DR}(t)$: Asset real energy demand during DR event, in kW.
- $D_{baseline}(t)$: Asset baseline energy demand without DR event, in kW.
- $C_{DR,fuel}(t)$: Asset real fuel consumption (for each type of fuel) during DR event, in kg/h.
- $C_{baseline,\,fuel}(t)$: Asset baseline fuel consumption (for each type of fuel) without DR event, in kg/h.
- $MIX_{source}(t)$: Proportion of the national electricity mix (index source corresponding to the production sources, as diesel, gas, coal, nuclear, hydropower, wind, solar, etc.).
- $EF_{source}$: Emission factors of national production sources and district heating supplier, in kg $CO_2$eq/kWh.
- $EF_{fuel}$: Emission factors of locally consumed fuel (for all different fuels), in kg $CO_2$/kg.

The only output datum is:

- $I_{CO_2,reduction}(\Delta t)$: Reduction of greenhouse gases emission (negative in case of emission increase), in kg $CO_2$.

The reduction of $CO_2$ emissions is taking into account the fuel, district heating, and electrical consumptions separately [28]:

$$I_{CO_2,reduction}(\Delta t) = \sum_{t\in\Delta t} \Delta I_{CO_2}(t) \tag{6}$$

With:

$$\begin{aligned}
\Delta I_{CO_2}(t) = &\sum_{source\in\{sources\}} \left(D_{DR,elec}(t) - D_{baseline,elec}(t)\right)MIX_{source}(t)EF_{source} \\
&+ \sum_{fuel\in\{fuels\}} (C_{DR,fuel}(t) - C_{baseline,fuel}(t))EF_{fuel} \\
&+ \left(D_{DR,distr\,heating}(t) - D_{baseline,distr\,heating}(t)\right)EF_{distr\,heating}
\end{aligned} \tag{7}$$

Index 'source' corresponds to the national production sources of electricity (i.e., Diesel, Gas, Coal, Nuclear, Wind, and Solar), whose proportions MIX are time varying.

The emissions factors (EF) for electricity sources are reported in kg $CO_2$eq/kWhelec and are based on life cycle analysis of the production sources. They are extracted from the Ecoinvent database (ECOINVENT, 2017) which is not only taking into account the production type of electricity, but also the national context of this production (e.g., the difference between French and Romanian nuclear power technologies). The electricity MIX for all countries can be gathered from the ENTSOE-E database (ENTSOE-E, 2017).

Index 'fuel' corresponds to the different fuels involved in the DR event (gas, diesel, wood, etc.).

Index 'distr heating' corresponds to the district heating energy factor. The related emission factor is specified in the adaptation to the Italian pilot site.

### 2.4.3. Economic Benefits

The economic gain corresponds to the overall benefit in national currency (£, €, etc.,) due to the DR implementation.

The needed measures and information are:

- $D_{DR}(t)$: Asset real energy demand during DR event, in kW.
- $D_{baseline}(t)$: Asset baseline energy demand without DR event, in kW.
- $S_{DR,elec}(t)$: Electricity selling during DR event, in kW.
- $S_{baseline,elec}(t)$: Electricity selling baseline without DR event, in kW.
- $C_{DR,fuel}(t)$: Asset real fuel consumption (for each type of fuel) during DR event, in m$^3$/h.
- $C_{baseline,\ fuel}(t)$: Asset baseline fuel consumption (for each type of fuel) without DR event, in m$^3$/h.
- $Pr_{elec}(t)$: Electricity sales tariff (bought from the grid), in national currency per kWh.
- $Pr_{elec,feedin}(t)$: Electricity feed-in tariff (sold to the grid), in national currency per kWh.
- $Pr_{fuel}$: Fuel tariff (for each type of fuel), in national currency per m$^3$.
- $Pr_{distr\ heating}$: District heating tariff, in national currency per kWh.
- $FR_{DR,util}$: Utilization payment of related DR program, in national currency or national currency per kW per hour.
- $FR_{DR,avail}$: Availability payment of related DR program, in national currency.

The only output datum is:

- $EG\ (\Delta t)$: Economic gain from DR scenario, in national currency.

The economic gain in calculated by summing the financial rewards and the energy and fuel expense variations:

$$EG(\Delta t) = \Delta FR(\Delta t) + \sum_{t \in \Delta t} \Delta Ex(t) \tag{8}$$

$\Delta Ex$ corresponds to the energy expenses variations (electricity, fuels, and district heating) [28]:

$$\begin{aligned} \Delta Ex(t) = \ & \left(D_{baseline,distr\ heating}(t) - D_{DR,distr\ heating}(t)\right)Pr_{distr\ heating} \\ & + \left(D_{baseline,elec}(t) - D_{DR,elec}(t)\right)Pr_{elec}(t) \\ & + \sum_{fuel \in \{fuels\}} (C_{baseline,fuel}(t) - C_{DR,fuel}(t))Pr_{fuel} \end{aligned} \tag{9}$$

$\Delta FR$ corresponds to the financial rewards variations, including electricity selling and specific incentives from the demand response programs (only for UK and FR pilot sites):

$$\begin{aligned} \Delta FR(\Delta t) = \ & FR_{DR,util} + FR_{DR,avail} \\ & + \sum_{t \in \Delta t} \left(S_{DR,\ elec}(t) - S_{baseline,\ elec}(t)\right)Pr_{elec,feedin} \end{aligned} \tag{10}$$

2.4.4. The Prediction Model and Baseline Calculations

The Exponentially Weighted Extended Recursive Least Square (EWE-RLS) algorithm based upon a standard Kalman filter is used to estimate the value of the process model parameters in real-time and is implemented and used to predict the energy consumption a day ahead. The derivation of the entire EW-RLS algorithm can be found in many previous works, and the main formulae for its implementation are [30,31]:

$$\hat{\beta}(t) = \hat{\beta}(t-1) + K(t)e(t) \tag{11}$$

$$e(t) = y(t) - x^T(t)\hat{\beta}(t-1) \tag{12}$$

$$K(t) = \frac{P(t-1)x(t)}{\lambda + x^T(t)P(t-1)x(t)} \tag{13}$$

$$P(t) = \frac{1}{\lambda}\Big[P(t-1) - K(t)x^T(t)P(t-1)\Big] \tag{14}$$

Where:

- $\beta(t)$ is the vector of parameters to be estimated.
- $y(t)$ is the current measurement which is to be regressed against.
- $x(t)$ is a vector of regression variables.
- $K(t)$ is the estimator gain vector (Kalman gain vector), $P(t)$ is the covariance matrix.
- $e(t)$ is the prior residual error and $\lambda$ is the forgetting factor.

In LEM implementation, the forgetting factor $e(t)$ set to 0.994.

## 3. Results and Discussion

### 3.1. LEM-Based Baseline Generation

As mentioned earlier, the LEM tool, along with the integrated prediction code, is used to generate the baseline for the meter files. These meters are located at different levels of asset, building, and whole campus. The baseline has been generated for all of these levels, but three meters located at different levels with a Meter Point Administration Number (MPAN) and respective meter names enclosed in bracket have been selected as follows: 1332152031 (Main site electrical meter), 1332152080 (Clarendon electric sub-meter), and 1332152041 (Phoenix main electric meter). A linear interpolation technique has been used to fill the gaps, if found, for the meters' data and adjust outliers that can significantly affect results. The work considers and recognizes weeky working days/holidays and the term time/season during the year based on the pilot site. During the baseline generation process, the effects of adding prediction model parameters corresponding to two weeks, three weeks, and four week regressive components have been investigated. A baseline has been generated a day ahead for these meter values and the MAPE measured at each step of the prediction horizon in each case for the meter datasets. The results are displayed in Figures 5–11 below. Figure 5, Figure 7, and Figure 9 show the actual meter values of energy consumption (kWh) used for baseline generation with the corresponding predicted values for the prediction time horizon. Based on the prediction result, MAPE is calculated for the predicted steps for individual meters as displayed in Figure 6, Figure 8, and Figure 10. We summarized the performance of the implemented EWE-RLS algorithm by averaging the MAPE values of predicted steps in each of the three meters as shown in Figure 11.

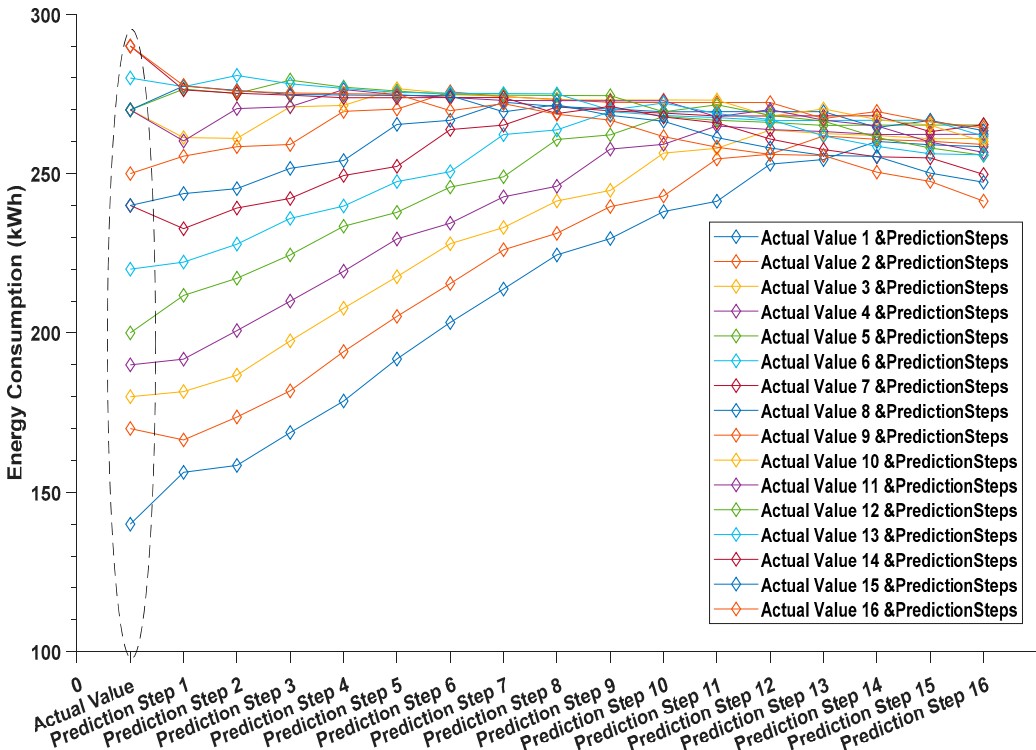

**Figure 5.** Meter 1332152031 values and 16 steps ahead energy prediction baseline.

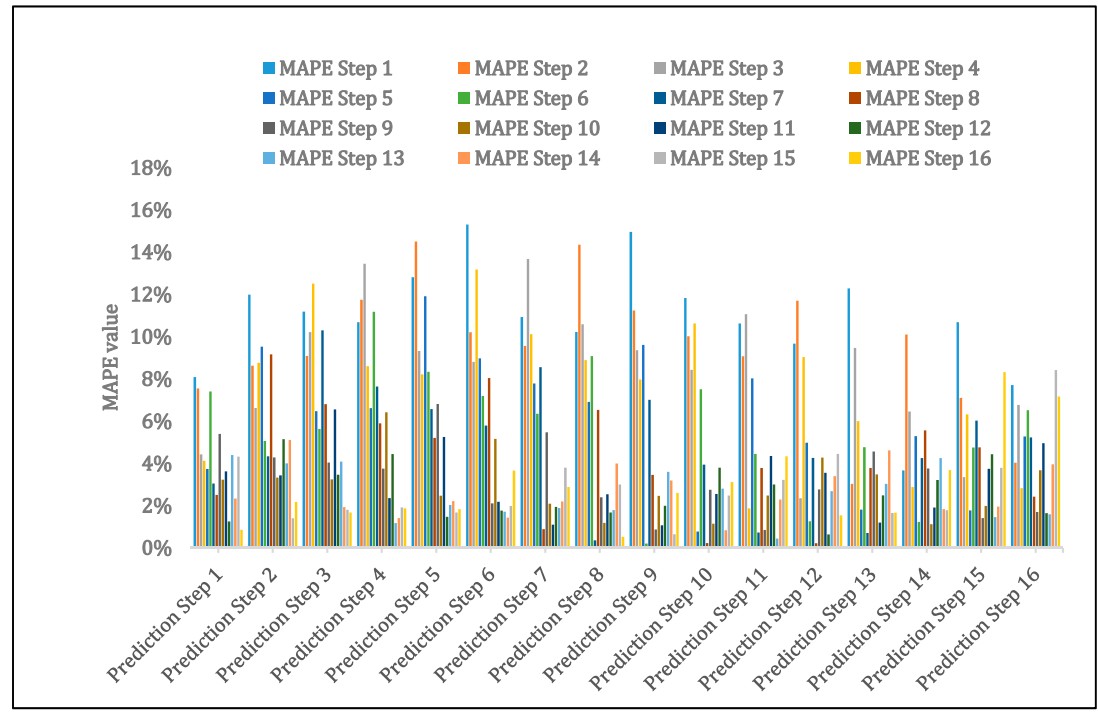

**Figure 6.** Meter 1332152031 Mean Absolute Percentage Error (MAPE) accuracy for 16 steps ahead.

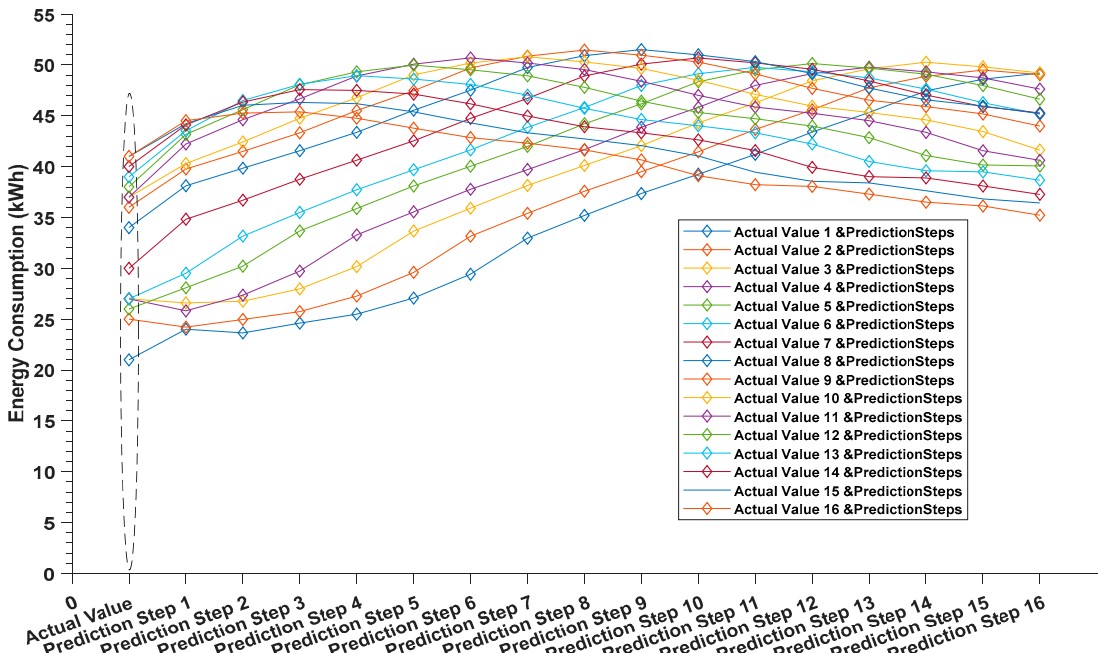

**Figure 7.** Meter 1332152080 values and 16 steps ahead energy prediction baseline.

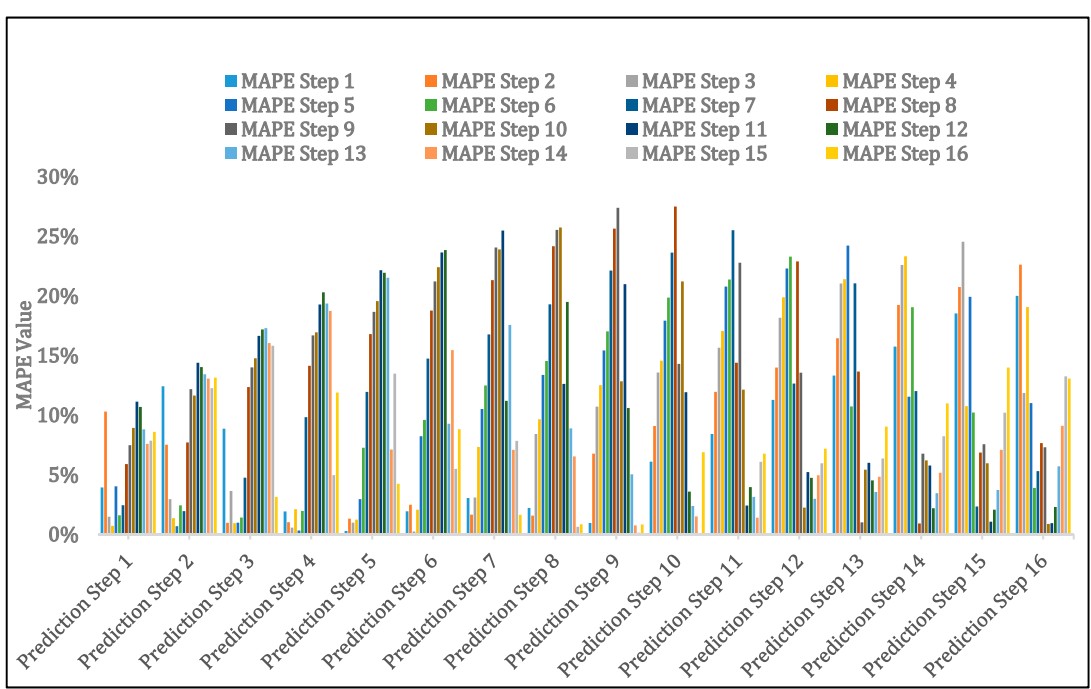

**Figure 8.** Meter 1332152080 Mean Absolute Percentage Error (MAPE) accuracy for 16 steps ahead.

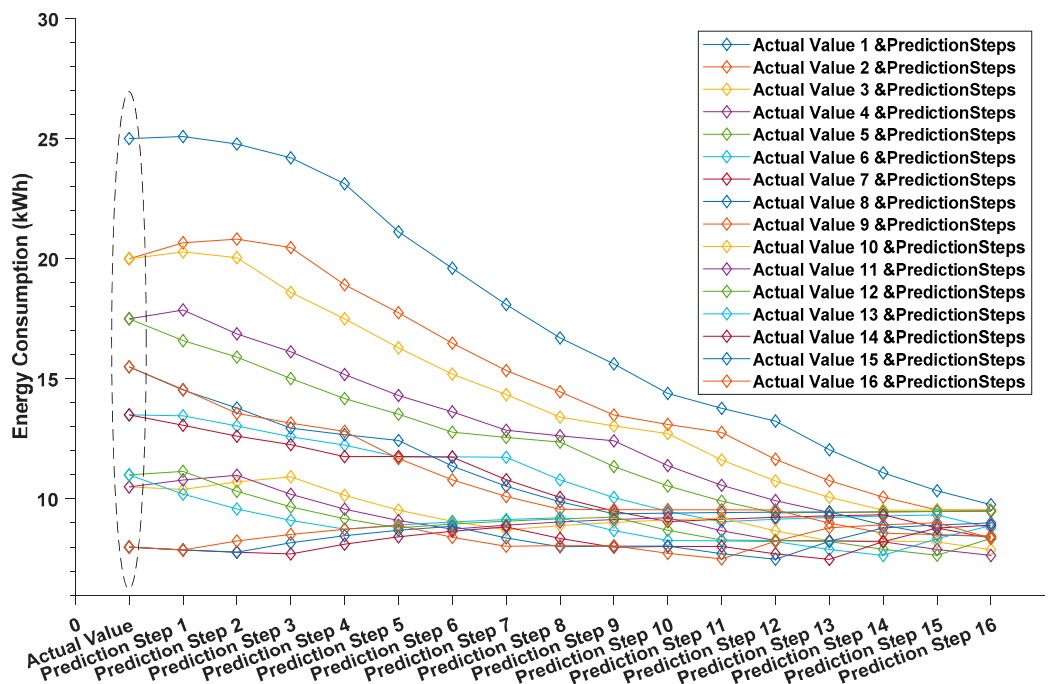

**Figure 9.** Meter 1332152041 values and 16 steps ahead energy prediction baseline.

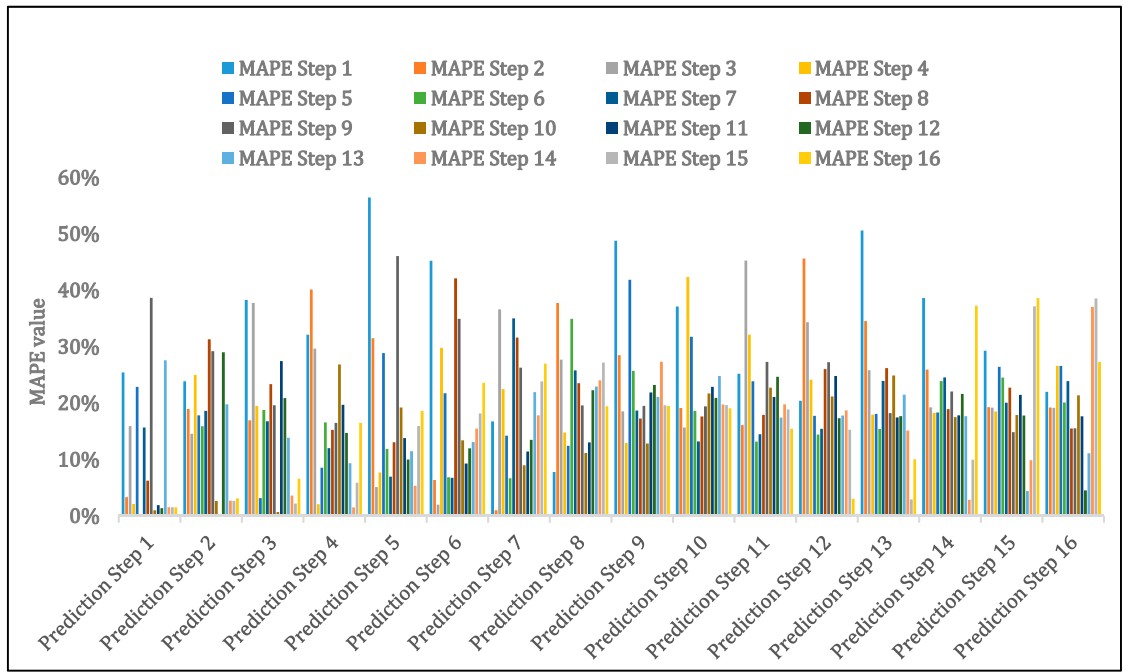

**Figure 10.** Meter 1332152041 Mean Absolute Percentage Error (MAPE) accuracy for 16 steps ahead.

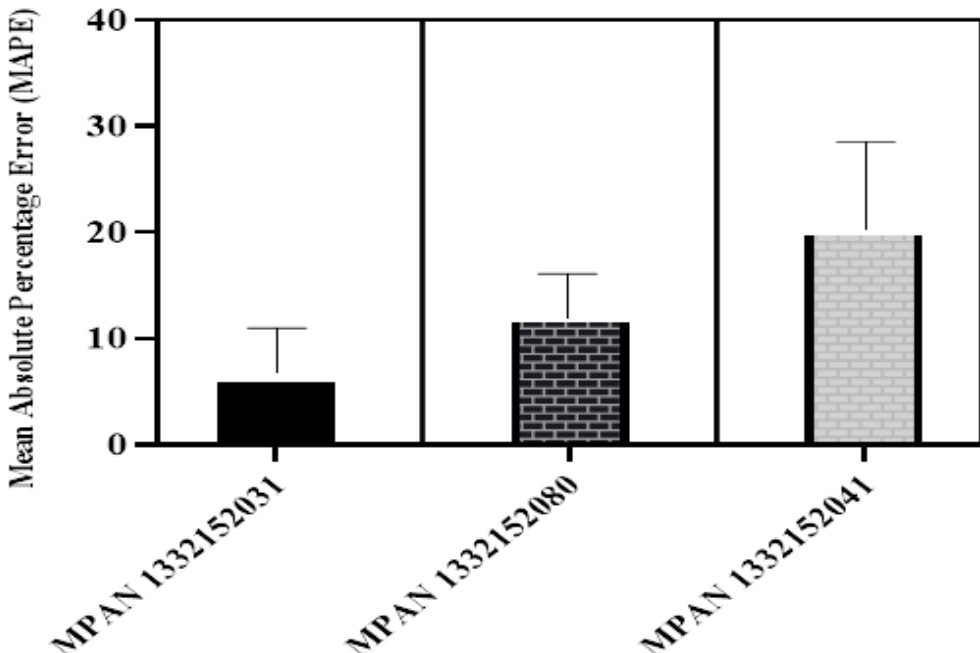

**Figure 11.** Average Mean Absolute Percentage Error (MAPE) accuracy for 16 steps ahead, 3 m.

The meter with the MPAN 1332152031 is the UK main site HV (in kWh) electricity meter for the whole Teesside University campus. It includes almost all the demand response scenarios in the UK of Scenario 1 (Short Term Operating Reserve, STOR in UK market), Scenario 2 (Demand Turn Up (DTU) in UK market), Scenario 3a (electric peak demand reduction-explicit), and Scenario 4 (frequency regulation/emergency load shedding).

Figures 5 and 6 illustrate a baseline of 16 steps ahead for meter 1332152031 along with their MAPE results. The 16 steps of 15 min each cover 4 h which overall is more than the time required for the longest demand response event scenario. It is clear from the results in both Figures 5 and 6 that the implemented EWE-RLS algorithm is able to predict the energy load very accurately with an average MAPE result of 5.1%, as clearly shown in Figure 11 for the above meter. The generated baseline shows also a very stable performance compared to the initial meter values.

The meter with the MPAN 1332152080 is the UK General Area Chillers electricity asset meter in the Clarendon Building at Teesside University (in kWh). It includes demand response UK Scenario 1 and Scenario 3a.

Figures 7 and 8 illustrate a baseline of 16 steps ahead for meter 1332152080 along with their MAPE results. It is clear from the results in both Figures 7 and 8 that the implemented EWE-RLS algorithm is able to predict the energy load accurately with an average MAPE results of 10.7% shown in Figure 11.

The meter with the MPAN 1332152041 is the UK electricity meter for the whole Phoenix Building at Teesside University (in kWh). It only includes the UK demand response Scenario 3a. Figures 9 and 10 illustrate a baseline of 16 steps ahead for meter 1332152041 along with their MAPE results. It is clear from the results in both Figures 7 and 8 that the implemented EWE-RLS algorithm is able to predict the energy load accurately with an average MAPE results of 19.6% shown in Figure 11. However, the prediction is clearly less accurate than other meters due to many reasons. These include the demand response scenarios involving this meter and the pattern and stability of the data generated by this meter, including the number of zeros and the missing data.

## 3.2. Baseline Validation and Assessment

The LEM baseline is used as the demand response baseline. To ensure that the prediction used for the generated baseline is appropriate and accurate enough to be used for this purpose, Siemens

Energy Bureau undertook an independent validation and assessment audit of the way in which the LEM baseline is calculated, generated, and its accuracy. Siemens assessed the data and detailed any assumptions, observations, and exceptions. Siemens reviewed the accuracy of the projections in all operational modes, reviewed the overall uncertainty against the magnitude of the DR events, and provided a summary of the findings against each meter in the UK pilot site. Siemens Energy Bureau has reported the validation and assessment audit for the UK pilot site, and this is an ongoing process for the rest of the pilot sites. Any adjustment to the baseline through LEM and the prediction algorithm depends on the outcome from the Siemens validation process results and the provided feedback and recommendations.

The following section provides more detail around each stage of the process. Two key documents inform the approach to this assessment. The International Performance Measurement and Verification Protocol (IPMVP) is the most commonly used and standard protocol for verifying energy savings and the Guideline How to Create a Consumption Baseline is a document generated by the S3C project, which sets out the principles of creating baselines in a demand response setting [30]. The applicable principles of the guide and applicable elements from IPMVP to this proposal are listed in detail in Table 1, and are as follows:

- Accuracy: a baseline should not systematically over or understate performance.
- Gaming: parties should not be able to influence the saving level by manipulating the baseline.
- Uncertainty: the savings need to be larger than twice the standard error all components including modelling and physical metering.
- Transparency: all elements of the baseline should be clearly stated and understood by all parties.
- Change Control: any alteration to the baseline calculations or meter should be agreed by all parties.

**Table 1.** Baseline calculations setting stages.

| Stage | Activity | Principle |
|---|---|---|
| Quality assurance of the data | To ensure that the base data are sound and query any concerns. | General |
| Assess data for patterns and assess accuracy | To understand whether the profiles respond to the cycles expected such as weekday/weekend and seasonality. The days selected to be tested for the accuracy of prediction will be drawn from each distinct profile. Any information about operational exceptions, local holidays, etc., will be taken into account if provided. | Accuracy |
| Assess baseline calculation | Review the baseline calculations to verify that it is clear, understood, and free of misconception. | Transparency |
| Review uncertainty of the model and meter against DR magnitude | The total uncertainty is calculated and compared to the magnitude of the DR event saving. Where the saving is not twice the uncertainty, this brings the validity of savings into question, which will be highlighted. | Uncertainty |
| Review change control | It is necessary for multiple parties to approve any change which could undermine savings validity and resulted in a significant risk. | Change |
| Review baseline application against DR event type | If the baseline uses a period adjacent to the DR event window to link the baseline profile to the actual profile, there is a risk that abnormal consumption during this period could distort the savings. By comparing the type of DR event and the baseline application method, the possibility of such distortion can be stated. | Gaming |

The following stages of analysis were undertaken.

The validation results from Siemens of the 3 months' data provided and their baselines confirm the accuracy of the results of the LEM tool. MAPE is found to be between 6%–16% when the data is adjusted, compared to 23%–31% without adjustments. Siemens' assessment results show that the overall MAPE results for all the meters located at different levels for 3 months is 22.93% [32]. Note that 20% hourly criteria in the case of IPMVP [33] has been used by Siemens to determine the length of the DR event for which the model can accurately predict. However, the validation results reveal that there is misalignment in the provided data. It also shows that there are a number of spikes which diverge from the weekly pattern that need to be tackled in the next baselines. This requires some changes to the LEM C++ code [30]. Pushing data forward can drastically improve the overall MAPE results and lead to a greater enhancement of the saving.

## 4. Conclusion

In this paper, accurate energy load forecasting baseline calculations have been developed and implemented to predict the short-term energy load. The proposed method is based on the Exponentially Weighted Extended Recursive Least Square (EWE-RLS) algorithm and a standard Kalman filter. The baselines have been generated and validated during and after running many demand response events in blocks of buildings. It is necessary to establish these baselines to represent, measure, and compare what the meter would have recorded in the absence of the demand response event, and also understand the impact of the event. Implementation results show that the generated baselines were accurate, and the error percentage found to be between 6%–16% when adjusting the data compared to 23%–31% without adjustments. The assessment and validation results show that the overall MAPE results for all the meters located at different levels for 3 months is 22.93%.

**Funding:** The DR BOB project (03/01/2016-28/02/2019) is co-funded by the EU's Horizon 2020 (H2020) innovation program under grant agreement No 696114.

**Conflicts of Interest:** The authors declare no conflict of interest.

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
