# Peer review of "Short-Term Prediction of Energy Consumption in Demand Response for Blocks of Buildings: DR-BoB Approach"

_buildings, doi:10.3390/buildings9100221_

Round 1

Reviewer 1 Report

The paper presents a new method to predict short-term energy consumption in buildings using the Exponentially Weighted Extended Recursive Least Square (EWE-RLS)  algorithm based upon a standard Kalman filter. A new software tool,  namely Local Energy Manager (LEM)  has been developed to implement the RLS algorithm and  predict the forecast for energy demand a day ahead with a regular meter frequency of quarter an  hour. The introduction is well written, and considers the references on this topic. For a comprehensive review, I suggest to add some recent research, for example https://doi.org/10.3390/en11102734 where an analysis of accuracy of determining of the seasonal heat demand in buildings based on short measurement periods is presented.

In my opinion, the text in Figures 2, 3, 5, 6, 7, 8, 9 and 10 is too small. Additionally, consider Figures 6, 8 and 10 in color, because they are unreadable in their present form.

Author Response

New reference was added as suggested by the reviewer. Figures have been enlarged and they are more readable now than the case before. It has been difficult to convert figures 6,8 and 10 in colour as the model produce the data in black and white images. However, i have enlarged the figure and they are more readable as the case before.

Reviewer 2 Report

The paper is dealing with the short-term prediction. The paper presents the results of a big EU project. A platform for data collecting and analysis was developed. The topic and the content of the paper are very good and relevant. However, the paper has to be improved.

General comments:

Please check the language in the entire paper. In general, the language is good, but the tenses are confusing. The work of the author has to be written in the past tense, while the published work has to be written in the present tense. Currently, there is a mix of many styles. Some sentences are not complete or are too long Please check the entire text. Figures have to be improved. The axis text and numbers have to be bigger.

Specific comments:

Line 14 – 16 – the sentence starting from “The EWE-RLS…” has to be checked and reformulated.

Line 27 – the first sentence has to be delete or reformulated. Currently, it does not bring so much.

Line 33 – “have been proposed” have to be deleted.

Line 110 – 111 – there is no context for the last sentence. The text before is very good, but it is not clear why the last sentence.

Line 139 – 140 – the sentence is not clear and complete.

Author Response

All English and tenses have been corrected in the paper as per the reviewer comments.

The specific comments highlighted by the reviewer were all corrected in the paper. 

Line 14-16: sentence was checked and corrected.

Line 27: first sentence has been deleted and the resent of the paragraph has been rewritten.

Line 33: sentence has been deleted

Line 110-111: sentence been re-structured.

Line 139-140: sentence has been re-structured.